# A Mechanical Evaluation of a Robot-Assisted Cutting Cornea Based on Force Response

**DOI:** 10.3390/mi14081634

**Published:** 2023-08-19

**Authors:** Qinran Zhang, Jingyu Zhao, Sikai Wang, Shijing Deng, Peng Su

**Affiliations:** 1School of Electromechanical Engineering, Beijing Information Science and Technology University, Beijing 100192, China; zhangqinran@bistu.edu.cn (Q.Z.); zhaojyy@163.com (J.Z.); skywang0306@foxmail.com (S.W.); 2Beijing Tongren Eye Center, Beijing Institute of Ophthalmology, Capital Medical University, Beijing 100730, China

**Keywords:** cornea, keratoplasty, trephination, robot-assisted cutting cornea, trephine force

## Abstract

The aim of this paper is to propose laws of trephine operation based on a robot-assisted cutting cornea in order to obtain better microsurgical effects for keratoplasty. Using a trephine robot integrated with a microforce sensor and a handheld trephine manipulator, robotic and manual experiments were performed, with porcine corneas as the test subjects. The effect of trephine operational parameters on the results reflected by the biomechanical response is discussed, and the parameters include linear velocity, rotating angle, and angular velocity. Using probability density functions, the distributions of the manual operational parameters show some randomness, and there is a large fluctuation in the trephine force during the experiments. The biomechanical response shows regular trends in the robotic experiments even under different parameters, and compared to manual trephination, the robot may perform the operation of trephine cornea cutting more stably. Under different operational parameters, the cutting force shows different trends, and the optimal initial parameters that result in better trephine effects can be obtained based on the trends. Based on this derived law, the operational parameters can be set in robotic trephination, and surgeons can also be specially trained to achieve a better microsurgical result.

## 1. Introduction

Trephination, an elaborate ophthalmic microsurgery, is one of the basic operations in keratoplasty, which can be divided into penetrating keratoplasty (PK) and deep anterior lamellar keratoplasty (DALK). PK is one of the most successful examples of tissue transplantation, and DALK is considered an alternative procedure to PK for corneal pathologies not affecting the endothelium and descemet membrane [1,2]. Some scholars have discussed the biomechanics of transplanted corneas after keratoplasty, and the interaction in the recipient–donor interface can influence the healing response [1,3]. Some endothelial cells have been lost in the trephination process, especially at the cutting edge [4], and this affects the effect of the surgery. However, modifying the surgical technique and instrumentation may be beneficial to minimizing this loss [5].

Nowadays, many surgeries of trephine cornea cutting are still manual operations in keratoplasty, which still strongly depend on the surgeons’ experience and technique. Evaluating the operation may provide insight into the contribution of trephination methods to decrease the likelihood of complications arising. A few researchers are exploring advanced techniques to achieve good trephine results, such as femtosecond laser technology, robotics, and so on [6,7]. The use of a femtosecond laser has some advantages, and it can prepare different configurations with a smooth cutting margin and a precise depth, but there are some drawbacks in clinical surgery. For example, a diseased cornea may affect laser energy, and the applications may be limited due to negative pressure suction in trephination. Robotics, the most similar method of manual operation, may overcome some shortcomings of manual operation, but the main application limitation arises from the need to ensure the reliability of micromanipulation in clinical microsurgery. Hu et al. [6] designed a robotic manipulator for trephine cornea cutting; one of the micromotors drove the trephine to punch the cornea linearly, and the other could realize the rotary motion based on the gear transmission. The mechanism is ingenious, but it did not consider that the rotation angle can be adjusted. Korff et al. [8] studied a semiautomatic trephination system, where a soft tissue preserving saw was combined with automatic adaptation of the cutting depth to protect the dura mater and reduce the bone gap. Taylor R et al. [9] described a surgical robotic device with a unique integrated sensor, and it may operate autonomously and evaluate the operation in real time using the state of tissues and disturbance information.

In these studies, effect of operational parameters on the biomechanical response is discussed thoroughly through the microsurgical process of trephination. The biomechanical response may be described by the forces and the torques [10], and some factors can affect the response (for example, the intraocular pressure (IOP) and velocity of trephine cornea cutting, etc.). Moreover, the trephine force and torque are complicated due to three-dimensional contact between the trephine and the cornea in the process of the trephine cutting the cornea, and the surgical operation involves two actions: linear punching and rotary cutting. The force *F* can be divided into three components, *F_x_*, *F_y_* and *F_z_*, and they are in the *X*-axis, *Y*-axis and *Z*-axis direction, respectively. Additionally, the torque *T* also contains three components of *T_x_*, *T_y_,* and *T_z_,* rotating about the *X*-axis and *Y*-axis and *Z*-axis, respectively [11,12]. Among them, the vertical force *F_z_,* which is perpendicular to the cutting plane, reflects the reaction force and the punching force from cut tissue, and vertical torque *T_z_* is one of the significant parameters used to describe biomechanical response, which determines the cutting force *F_c_* in the process [12]. In the experiments, the trephine force can be simplified as two components of the vertical force *F_z_* and the cutting force *F_c_* [13].

Trephine operations are important for the surgical effectiveness of keratoplasty, and should attract considerably more attention. This paper shows manual trephination using a series of experiments on porcine corneas. Firstly, the test systems of manual trephination experiments and robotic experiments were built, and the robotic system carried out the trephine operation stably. Secondly, groups of manual experiments of trephine cornea cutting were performed on porcine corneas, and the operation law for manual experiments was obtained based on a probability density function, which describes the three parameters of the linear velocity, the rotating angle, and the angular velocity. Analyzing the trephine force that can be used as a basis for evaluating the surgical effect, the data were calculated to obtain the expected values of the parameters. Then, groups of the manual and robotic experiments of trephine cornea cutting were performed, and the optimal parameters of trephination were obtained by varying the cutting force under different initial conditions in robotic experiments, and the advantages of robotic trephination were also shown by the experiments. This paper evaluates the mechanical response of trephine cornea cutting under different operational parameters based on the force response of the manual and robotic experiments, tries to reveal some of the laws of microsurgery based on a performance evaluation of trephine operations, and promotes the progress of technology of trephine cornea cutting for keratoplasty.

## 2. Materials and Methods

In the experiments, manual and robotic trephination experiments were performed using a disposable trephine with a diameter of 7.5 mm, and the test subjects chosen were fresh porcine eyeballs. Although there are some differences between the human cornea and the porcine cornea [14], the porcine cornea is applicable to discussions of the effects of trephine operations on biomechanics, because they have enough similarities in their mechanical response. In order to reduce the errors in many experiments, the eyeballs were injected with normal saline to make the intraocular pressure value normal, i.e., 20.6 mmHg, as measured using a dynamic contour tonometer [15].

### 2.1. Manual Experiments

A handheld trephine system comprises a handheld trephine manipulator integrated with the microforce sensor (i.e., NANO 17, ATI company, Charlotte, NC, USA [16]), a high-definition (HD) image acquisition system (i.e., NEX-VG30EH, SONY company, Tokyo, Japan), and a data-processing system, as shown in Figure 1. Considering the significant randomness in manual experiments due to different surgeons with different surgical experiences and styles, ten surgeons performed the experiments using the handheld manipulator, carrying out PK-trephination and DALK-trephination on 50 eyeballs each.

In the experiments, the trephine force data are measured by the sensor, which is an expressive form of biomechanical response, and can be used as for the basis for judging results. The operative actions of manual trephination are recorded using a high-definition camera, the image data are collected using a data-processing system, and the effective marks of the image are extracted to match with the data of the forces. According to the data analysis, the operational parameters of the manual experiments are obtained, including linear velocity, rotating angle, and angular velocity. Finally, the distributions and the expected values can be calculated based on probability theory.

### 2.2. Robotic Experiments

In many surgeries, robots can assist surgeons in undertaking some tasks, such as suturing and cutting; they require smart control, the ability to sense the force of the contact between instrument and tissue, and the ability to predict the outcome of operations [6]. An experimental system for robotic trephination has been designed, which comprises a surgical robot, a trephine manipulator, a data-processing system, and a control system, as shown in Figure 2. The surgical robot, which has been described in detail in a previous paper, can adjust the position and posture based on the movements of three rotating joints [17]. The manipulator can perform a trephine operation, as the surgeon contains a linear punching motion and a rotary cutting motion. This integrates two motors; one stepper motor drives one precision screw (300 μm screw pitch, 100 mm effective length) to achieve rectilinear motion. The other motor achieves forward and reverse rotational motion. The circumferential motion is realized by controlling the step motor with the pulse number after subdivision, and the motions can be controlled independently based on the control system, which integrates the STM32 Microcontroller Unit. The encoder monitors the motion of the disposable trephine and the output motion parameters to avoid operation errors caused by lost motor steps or out-of-control speed. It is important for a successful experiment that the rotational cutting is a concentric motion, and in the manipulator, the bearing and the holder are used reasonably to ensure that the trephine can obtain a stable rotational movement from the motor.

Because the trephine depth is not discussed in this paper, and is a key aspect of DALK, the experimental system for robotic trephination performs three groups of PK-trephination experiments only. In each group, 50 experiments are carried out, and one of the three expected parameters obtained in the manual experiments is varied, while the other two are kept constant. Through mathematical analysis of the experimental results, this study tries to find the variations of trephine forces under different initial parameters.

## 3. Results

Due to the possibility of greater randomness in manual operations, the study analyses the data from physician operations by means of a probability density function to obtain probability graphs for the samples. In probability theory, a probability density function (PDF) or the density of a continuous random variable is a function that describes the relative likelihood of this random variable taking on a given value. The horizontal coordinate of the peak value in the graph is the mathematical expected value of the random variable, and the vertical value is the probability density value. PDFs can give the expected value, which is more comprehensive than the mean value because it can reflect the distribution of the data.

### 3.1. Statistics and Analysis of Operational Parameters on Manual Experiments

There is randomness attributed to the surgeon’s experience, technique, and involuntary tremble in microsurgery, so the data of manual experiments are analyzed by building a PDF. The statistics of the operational parameters of manual trephine surgery are shown in Table 1, where the parameters calculated are the mean values in each experiment. The probability graphics of the parameters are shown in Figure 3.

Firstly, the linear velocity is related to the vertical force *F_z_* applied by the surgeon, which is the main factor that affects the operation time. In DALK-trephination, the surgeons need to determine the trephine depth precisely, so that they have to perform a more careful operation. Hence, the velocity is much smaller than in PK-trephination, and the distribution of the parameters is more concentrated, as shown in Figure 3a. Secondly, there is a smaller rotating angle in PK-trephination, which is about 75 percent of the angle in DALK-trephination, as shown in Figure 3b. The performance reflects that the surgeon chooses a special treatment for each type of surgery. Finally, the angular velocity is related to the habits of the surgeon, which are formed by long-term performance of the operation; if there is no special training, it is difficult to change this with technology. Although there is an obvious difference in the rotating angle, Figure 3c shows that the angular velocities of the two types are close, including the expected value and the distribution.

In general, the trephine force is affected by all three parameters. The operational parameters for PK-trephination have a lower probability density, and the surgeon seems to perform a more casual operation for PK-trephination than for DALK-trephination.

### 3.2. Statistics and Analysis of Trephine Force

The statistics of trephine force are shown in Table 2, where the value of the vertical force *F_z_* is the maximum in each experiment, and the cutting force *F_c_* is the maximum of the peak-to-peak value, which is calculated by dividing the vertical torque *T_z_* by the length of the arm applying the force *l* (*l* = 37.5 mm). The PDF of the trephine force is shown in Figure 4.

In PK-trephination, the vertical force *F_z_* is greater than the value in DALK-trephination, because the reaction force is larger, in large part caused by the extrusion of the eyeball, and the reaction force increases with trephine depth, as shown in Figure 4a. However, in Figure 4b, the cutting force *F_c_* is smaller in the process of PK-trephine surgery, and it is not affected by the trephine depth. Overall, in agreement with Figure 3 above, the same result arises: the distributions of the trephine force are more concentrated in DALK-trephination.

### 3.3. The Variation in the Trephine Force under Different Operational Parameters

Randomness is inevitable in manual microsurgery, so it is necessary to explore the impact of the operational parameters on the results of the surgery. As a typical representative of biomaterials, cornea has two characteristics: superelasticity and viscoelasticity, and the cornea on the eyeball is bound by the sclera, which is also a biological material. These factors together lead to the cutting force being affected by linear velocity, the rotating angle, and angular velocity, etc. In each set of robotic experiments, two of the desired values of the operating parameters obtained in manual surgery were used as fixed values, and the value of the other parameter varied from small to large, for 50 trephine cutting experiments each. Based on the series of robotic experimental data, the variations of trephine forces are discussed under operational parameters including linear velocity, rotating angle, and angular velocity. The findings suggest that there are similar variations between the vertical force *F_z_* and the cutting force *F_c_*, so the section will show the change in the force Fc under different initial parameters because it reflects the law without interference from the reaction force [12], as shown in Figure 5.

Firstly, the rotating angle and the angular velocity are constants, and the variation is discussed under different linear velocities. Based on the expected parameters derived from the manual experiments, the constants are set, i.e., [*θ*, *ω*] = [1.00, 9.87]. The force *F_c_* generally increases with linear velocity, and when the velocity is in the interval AB (i.e., 75 μm/s > *v* > 250 μm/s), the increasing trend is most obvious, as shown in Figure 5a. Secondly, the variation of the force is discussed under different rotating angles, and the other two parameters are set to be constants derived from Table 1, i.e., [*ν*, *ω*] = [34.52, 9.87]. As shown in Figure 5b, the force *F_c_* shows fluctuations, which means the force shows a clear upward tendency before point A, and then declines significantly to reach the trough point B, and after point B, the force slowly increases again. In the curve, there are two troughs, point O and point B, but point B should be the optimal parameter, because the angle is small enough to avoid an unstable operation. Thirdly, the variation is discussed under different angular velocities, where the constants are set to [*ν*, *θ*] = [34.52, 1.00]. The force *F_c_* usually decreases with angular velocity, and before point A, where the velocity is 15 rad/min, the decreasing trend is at its most obvious, as shown in Figure 5c.

The result of each experiment can show the variation in the trephine force in real time, rather than the average of many results, so an experiment is selected for the analysis of the results. The manual experiment and the robotic experiment will be analyzed, respectively.

### 3.4. The Results of the Manual Experiments

The experiments whose initial parameters are closest to the expected values are chosen, including a PK-trephination experiment and a DALK-trephination experiment. The results of the two experiments are shown in Figure 6.

Analyzing the manual experiments, two experiments are selected whose initial parameters are closest to the expected values in Table 1, where the initial parameters of PK-trephination [*ν*, *θ*, *ω*] = [34.50, 1.02, 9.85], the parameters experiment of DALK-trephination [*ν*, *θ*, *ω*] = [21.70, 1.27, 9.75], and the results of the experiments are shown in Figure 6. In Figure 6a,b, the dotted lines and the solid lines represent the results from PK-trephination and DALK-trephination, respectively.

In Figure 6a, DALK-trephination is carried out, which starts at point A and stops at Point B; the trephine depth is about 50 percent of the corneal thickness after 22.55 s. Vertical force *F_z_* can clearly show an increasing fluctuation caused by rotation and significant volatility, and the force reaches the maximum at point C, at about 16.48 × 10^−2^ N. In the first period, the peak is small, because the surgeon needs to adapt by using trial and error. Furthermore, there are larger fluctuations where the upward trend of the peaks is obvious and the troughs are close to zero, because the vertical velocity is so small that the surgeons find it difficult to exert a sustained extrusion force. In the PK-trephination, the force also shows an increasing fluctuation and uncertain volatility, but the maximum of *F_z_* is greater due to deeper depth, and the troughs can be kept in the process of the experiment because the depth is not considered. In Figure 6b, the curve of the cutting force *F_c_* shows an approximate periodicity, with almost always some volatility in each period due to the instability of the operation. It is necessary to determine the reasonable operational parameters in manual trephination to reduce unknown volatility.

### 3.5. The Results of the Robotic Experiments

Based on the variation of the cutting force under different operational parameters, the optimal initial parameters of the robotic experiment are chosen as [*ν*, *θ*, *ω*] = [20.00, 1.30, 33.50], and the experimental results are shown in Figure 7.

In Figure 7a,b, the effective range of the experiment is from point A to point B. However, the linear velocity is so small that it leads to an increased surgery time (about 90 s), and the number of rotations is so large that the result shows concentrated curves, because the angular velocity is fast enough, but the rotating angle is relatively small. In Figure 7a, the vertical force shows a growing fluctuation, and the trephine always presses the cornea during the experiment. The force *F_z_* reached the maximum at point C at about 14.92 × 10^−2^ N, and the punching force *F_p_* reaches a maximum of 2.30 × 10^−2^ N. Compared with Figure 6b, Figure 7b shows a change in the law, leading to a much more stable behavior, and the force *F_c_* has an obvious tendency to increase, with a value less than 2.40 × 10^−2^ N.

## 4. Discussion

The manual and robotic experiments of trephine cornea cutting were performed on porcine cornea, and this paper tries to reveal some of the laws of microsurgery based on a performance evaluation of trephine operations. Some scholars have paid close attention to corneal biomechanical changes after operations, and discussed the effect of the interface and the edge on the healing time of wounds [17,18]. However, trephine forces should also attract more attention, and be used as a basis for judging surgical effects and helping surgeons perceive this process; they may be used as a reference to make the surgery safer [10,12,19].

Different operational parameters, including linear velocity, rotating angle and angular velocity, etc., can lead to different trephine forces. Using a probability density function, the effect of different operational parameters on manual experiments of PK-trephination and DALK-trephination is discussed, and this shows the habits of the surgeon formed by long-term operations. It also shows that these habits are difficult to change without special training. The studies show some randomness in the process, and that there is a significant difference in the linear velocity and rotating angle, particularly during the two types of trephination. The trephine force of manual experiments, shown as the vertical force and the cutting force, is analyzed using the same probability theory. The trephine force is the most significant parameter when evaluating the surgical effect. The distribution of the trephine force is more concentrated in DALK-trephination due to the need for a careful operation. The results of manual trephination are produced by two experiments whose initial parameters are closest to the expected values of the two types. In general, there is some uncertainty and volatility, as seen in the large fluctuations in the curves of the trephine force.

A surgical robot can assist an operator to achieve a highly skilled result, and a unique sensing and control approach is necessary to perform delicate surgery when using such devices [20]. This paper describes a robotic system that can carry out the trephine operation stably. A series of experiments were performed wherein the behavior of robotic trephination using the manipulator was analyzed. Under different operational parameters, the cutting force shows different trends; for example, it increases with linear velocity, and decreases with angular velocity. It verifies the law’s prediction that better trephine effects can be obtained under optimal initial parameters. Based on the law, some reasonable parameters can be determined, which can be used to guide the surgeons.

By comparing the results of manual trephination experiments and robotic ones, including the change laws of the trephine force, the effectiveness of the manipulator has been evaluated. Robotic trephination has shown some advantages in that it can generate a small and stable cutting force, and it can increase the similarity of the donor and the recipient so the robot may be developed to produce more accurate results, which helps the biomechanical properties of the cornea as a whole [5].

From the evaluation of robotic trephination, it was found that the operational parameters affect the surgical effects, such as the recovery time and astigmatism [3]. It is important to determine reasonable operational parameters in manual trephination, and it is necessary to train surgeons according to these parameters. Although the effects of the operational parameters on the biomechanics of trephine cornea cutting have been discussed, the laws and the robots applied in clinical surgery need to be further elucidated, and more in-depth statistical analysis is needed, considering standard deviations, errors, etc. Additionally, there are more factors to be considered, including tests in clinical practice, control of trephine depth, and the initial position of the trephine. We believe that the research carried out herein and further research in this area will promote the progress of technology of trephine cornea cutting for keratoplasty, including manual skill, assisted robots, and so on.

## Figures and Tables

**Figure 1 micromachines-14-01634-f001:**
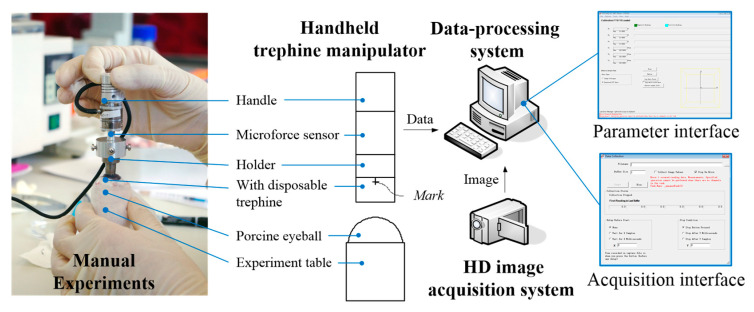
The experimental system of manual trephination.

**Figure 2 micromachines-14-01634-f002:**
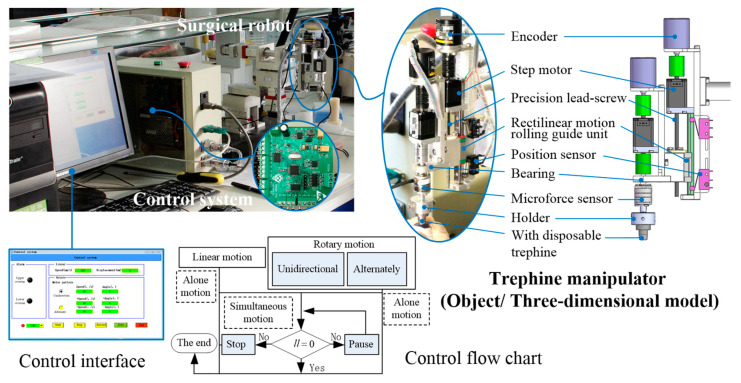
The experiment system of robotic trephination.

**Figure 3 micromachines-14-01634-f003:**
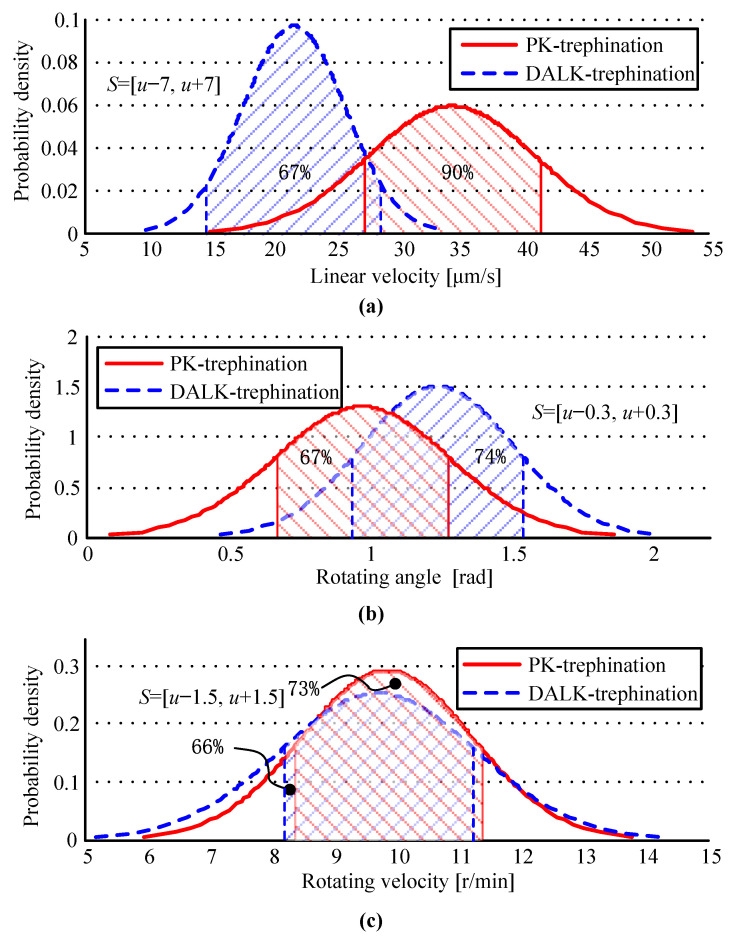
Probability graphics of operational parameters for manual experiments. The percentage, i.e., probability density *p*, is the probability value of a random variable being in the sample interval *S*. The figures (**a**–**c**) describe the values of the probability density for the linear velocity, rotating angle, the angular velocity, respectively.

**Figure 4 micromachines-14-01634-f004:**
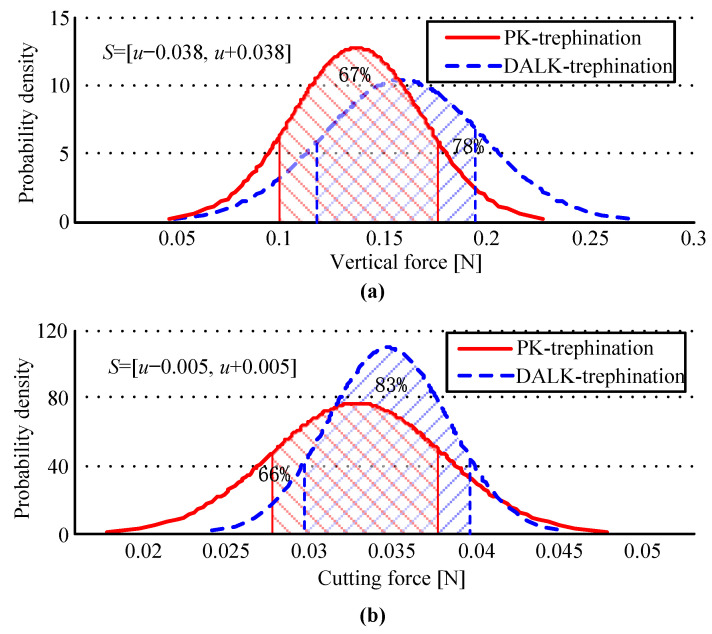
Probability graphics of the trephine force for manual experiments. Figure (**a**,**b**) describe the PDF for the vertical force *F_z_* and the cutting force *F_c_*, respectively.

**Figure 5 micromachines-14-01634-f005:**
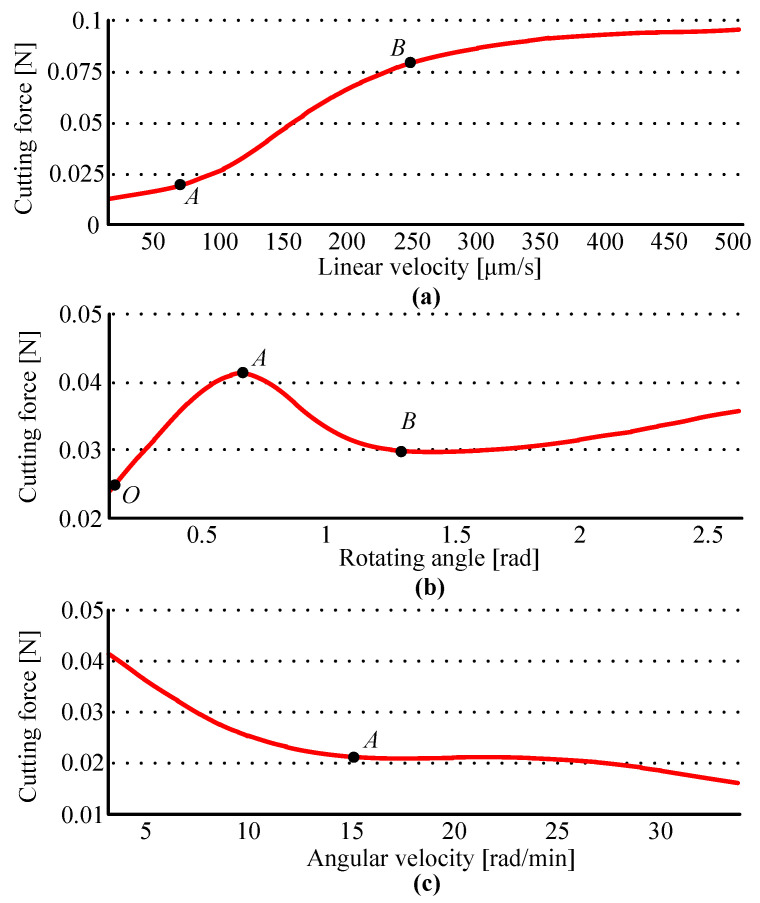
The variation of the cutting force *F_c_* under different operational parameters. Figures (**a**–**c**) are the results obtained by changing the linear velocity, rotating angle, and angular velocity, respectively.

**Figure 6 micromachines-14-01634-f006:**
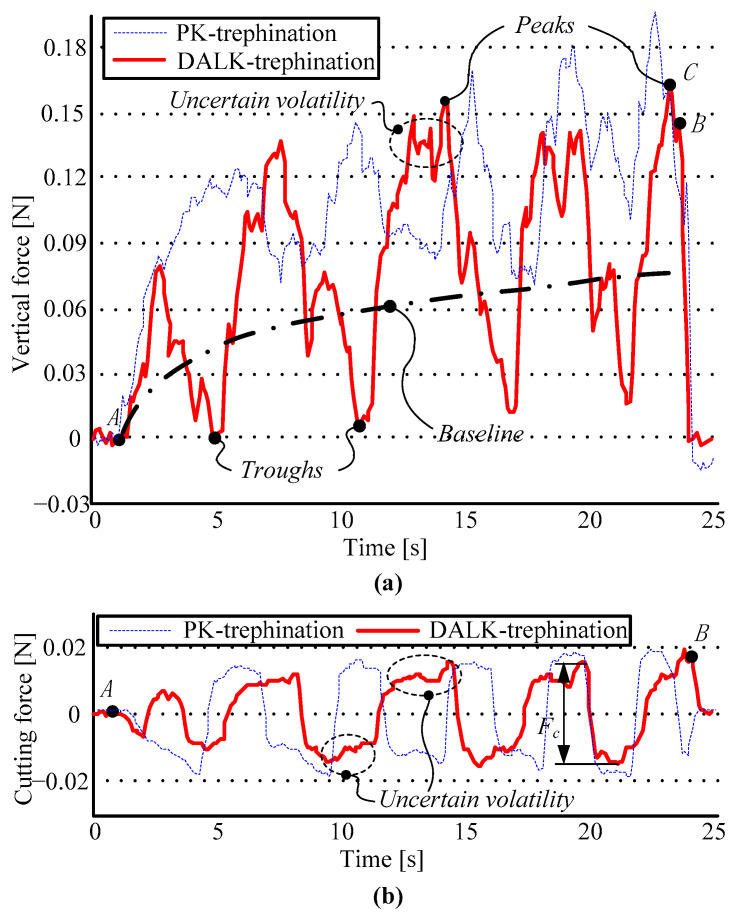
The results of the manual experiment under expected parameters. Figures (**a**,**b**) show the curves of vertical force *F_z_* and cutting force *F_c_*.

**Figure 7 micromachines-14-01634-f007:**
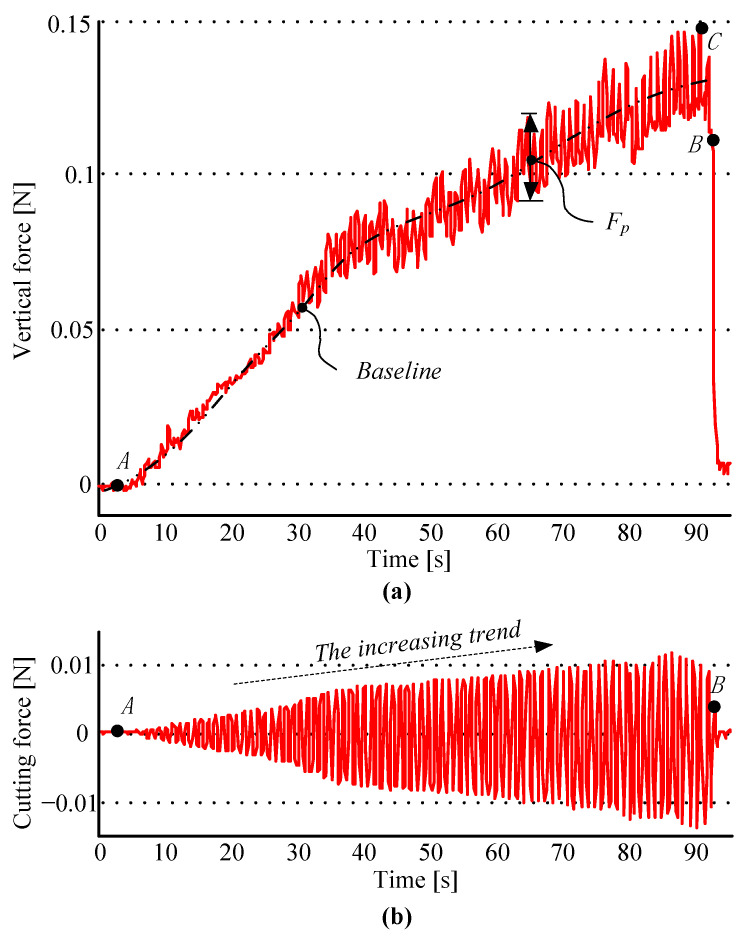
The results of robotic experiments under optimal initial parameters. Figures (**a**,**b**) show the curves of vertical force *F_z_* and cutting force *F_c_*.

**Table 1 micromachines-14-01634-t001:** The statistics of operational parameters for manual experiments.

Action Parameter	The Type	The Mean *m*	Expected Value *u*
Linear velocity *v* (μm/s)	PK	34.31	34.52
	DALK	21.70	21.75
Rotating angle *ϴ* (rad)	PK	0.97	1.00
	DALK	1.24	1.25
Angular velocity *ω* (rad/min)	PK	9.80	9.87
	DALK	9.67	9.72

**Table 2 micromachines-14-01634-t002:** The statistics of the trephine force for manual experiments.

Parameters	The Tye	The Mean *m* (×10^−2^)	Expected Value *u* (×10^−2^)
Vertical force *F_z_* (N)	PK	15.90	15.50
	DALK	13.70	13.80
Cutting force *F_c_* (N)	PK	3.31	3.30
	DALK	3.47	3.49

## Data Availability

Not applicable.

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
