# Peer review of "A Mechanical Evaluation of a Robot-Assisted Cutting Cornea Based on Force Response"

_micromachines, 2023, doi:10.3390/mi14081634_

Round 1

Reviewer 1 Report

The paper is about mechanical evaluation of robot-assisted cutting cornea. It starts with some manual experiments followed by robot-assisted experiments. The paper should be improved in many ways to be publishable. Here are my suggestions/comments:

1. The introduction of the paper can be improved significantly by providing newer references. many references and very old.

2. For the manual experiments, authors should explain how this is different from what has been done in ref [10].

3. Figure 1 has to be improved significantly. Every data-processing system has a computer and a user interface. The picture of use interface is not clear. How that image helps readers to get information about the system? The control system picture that is provided just shows a PCB board with no information about how it works. The interface for the force sensor is not shown at all. The design of hand-held manipulator should be shown in exploded view. 

4. For manual experiments, the force measurements are noisy. It is not explained how this problem was considered and fixed.

5. There is no information on the HD image acquisition system and just a picture of a camera on a tripod is shown. 

6. For the robotic experiments, again, the design of the system should be shown in exploded view and explained. How the controller controls the motion is not clear (a block diagram of how different signals are related to each other, such as position sensor readings, encoder readings, force sensor readings, stepper motor control signal, etc.).

7. For the statistical analysis, just the mean value is provided. More in-dept statistical analysis is required providing standard deviation, error, etc.

8. A discussion on Figure 5 is necessary to explain why these figures are expected under different conditions. For example, why changing the rotating angle causes the cutting force to change that way.

9. It is not explained how these results will be generalizable to actual surgeries. For example, how the angle of approach will change these results?  

There are some minor typos in English that should be fixed. For example, line 238, "the force also show" to "the force also shows". The paper should be proof-read for these.

Author Response

Thank you for your comments! This is our reply, please check.

Reviewer 2 Report

1. The technical novelty and main contributions of this article shall be presented clearly in the Introduction section. The current version didn't reflect well of these aspects. The reviewer recommend the authors modify relevant contents to strengthen this point.

2. The reviewer understand that the experiments were repeated for 50 times for each scenario. Please clarify how was the recorded data being processed that demonstrated  (Figure 4 -7) in this article.

3. As this article obtained results from animals experiments, which involves the matter of animal ethics. The reviewer didn't see any ethical approval evidence for the experiment presented in this article. Please provide necessary supporting information on this point.

The authors shall improve the quality of the language to avoid confusion. Take the sentence "A few researchers are exploring advanced techniques to achieve good trephine results, such as femtosecond laser technology, robotics, and so on [6,7]. " within line 40-42 on page 1 for example. Obviously, the authors are talking about "research" rather than "researcher". Similar expression found in sentence "The operation law on manual experiments is obtained based on a probability density function, which describes three parameters of the linear velocity, the rotating angle, and the angular velocity [10]." within line 74-76 on Page 2. The reviewer guess the authors means the "linear velocity, rotating angle, and angular velocity" are the three parameters, but the expression is lack of clarity. The authors are highly recommended to proofread the entire paper.

Author Response

(The authors gave the same response as above.)

Round 2

Reviewer 1 Report

The authors addressed my concerns.

The quality of English is fine. Minor spell checking is required.